The Global Museum: natural history collections and the future of evolutionary science and public education

Bakker Freek T. freek.bakker@wur.nl 1
Antonelli Alexandre 2
Clarke Julia A. 3
Cook Joseph A. 4
Edwards Scott V. 5 6
Ericson Per G.P. 7
Faurby Søren 8
Ferrand Nuno 9
Gelang Magnus 10 12
Gillespie Rosemary G. 11
Irestedt Martin 7
Lundin Kennet 10 12
Larsson Ellen 8 12
Matos-Maraví Pável 13
Müller Johannes 14
von Proschwitz Ted 10 12
Roderick George K. 11
Schliep Alexander 15
Wahlberg Niklas 16
Wiedenhoeft John 15
Källersjö Mari 12 17
1 Biosystematics Group, Wageningen University & Research , Wageningen , The Netherlands
2 Department of Science, Royal Botanic Gardens, Kew , Richmond , United Kingdom
3 Jackson School of Geosciences, University of Texas at Austin , Austin , TX , United States of America
4 Museum of Southwestern Biology, Department of Biology, University of New Mexico , Albuquerque , NM , United States of America
5 Department of Organismic and Evolutionary Biology, Museum of Comparative Zoology, Harvard University , Cambridge , MA , United States of America
6 Gothenburg Centre for Advanced Studies in Science and Technology, Chalmers University of Technology and University of Gothenburg , Göteborg , Sweden
7 Department of Bioinformatics and Genetics, Swedish Museum of Natural History , Stockholm , Sweden
8 Department of Biological and Environmental Sciences, Gothenburg Global Biodiversity Centre, University of Gothenburg , Göteborg , Sweden
9 Museu de História Natural e da Ciência, Universidade do Porto , Porto , Portugal
10 Department of Zoology, Gothenburg Natural History Museum , Göteborg , Sweden
11 Essig Museum of Entomology, Department of Environmental Science, Policy and Management, University of California, Berkeley , Berkeley , CA , United States of America
12 Gothenburg Global Biodiversity Centre, University of Gothenburg , Göteborg , Sweden
13 Biology Centre of the Czech Academy of Sciences, Institute of Entomology , České Budějovice , Czechia
14 Leibniz-Institut für Evolutions- und Biodiversitätsforschung, Museum für Naturkunde , Berlin , Germany
15 Department of Computer Science and Engineering, University of Gothenburg , Göteborg , Sweden
16 Department of Biology, Lund University , Lund , Sweden
17 Gothenburg Botanical Garden , Göteborg , Sweden
Cardoso da Silva José Maria
Electronic publication date: 2020 Jan 28
Publication date: 2020
Volume: 8
Electronic Location ID: e8225
Received 2019 Apr 18; Accepted 2019 Nov 15
Copyright: ©2020 Bakker et al.
Copyright year: 2020
Copyright holder: Bakker et al.
License: This is an open access article distributed under the terms of the Creative Commons Attribution License, which permits unrestricted use, distribution, reproduction and adaptation in any medium and for any purpose provided that it is properly attributed. For attribution, the original author(s), title, publication source (PeerJ) and either DOI or URL of the article must be cited.
License URL: https://creativecommons.org/licenses/by/4.0/

Keywords: Field education, Specimens, Transcriptomics, Collections, Epigenomics, Innovation-incubator, Global museum, Natural history, Place-based

Funding: Swedish Research Council B0569601 European Research Council under the European Union’s Seventh Framework Programme FP/2007-2013 Swedish Foundation for Strategic Research Wallenberg Academy Fellowship Faculty of Sciences at the University of Gothenburg Wenner-Gren Foundations David Rockefeller Center for Latin American Studies at Harvard University Marie Sklodowska-Curie research fellowship MARIPOSAS-704035 PPLZ programme of the Czech Academy of Sciences L200961951 Wetmore Colles Fund of the Museum of Comparative Zoology, Harvard University This work was supported by the Swedish Research Council (B0569601), the European Research Council under the European Union’s Seventh Framework Programme (FP/2007-2013, ERC Grant Agreement 331024), the Swedish Foundation for Strategic Research, a Wallenberg Academy Fellowship, the Faculty of Sciences at the University of Gothenburg, the Wenner-Gren Foundations, and the David Rockefeller Center for Latin American Studies at Harvard University to Alexandre Antonelli; the Marie Sklodowska-Curie research fellowship (European Commission, project MARIPOSAS-704035) and the PPLZ programme of the Czech Academy of Sciences (grant L200961951) to Pável Matos-Maraví. The publication fees for this article were paid by a grant from the Wetmore Colles Fund of the Museum of Comparative Zoology, Harvard University. The funders had no role in study design, data collection and analysis, decision to publish, or preparation of the manuscript.

==============================
Natural history museums are unique spaces for interdisciplinary research and educational innovation. Through extensive exhibits and public programming and by hosting rich communities of amateurs, students, and researchers at all stages of their careers, they can provide a place-based window to focus on integration of science and discovery, as well as a locus for community engagement. At the same time, like a synthesis radio telescope, when joined together through emerging digital resources, the global community of museums (the ‘Global Museum’) is more than the sum of its parts, allowing insights and answers to diverse biological, environmental, and societal questions at the global scale, across eons of time, and spanning vast diversity across the Tree of Life. We argue that, whereas natural history collections and museums began with a focus on describing the diversity and peculiarities of species on Earth, they are now increasingly leveraged in new ways that significantly expand their impact and relevance. These new directions include the possibility to ask new, often interdisciplinary questions in basic and applied science, such as in biomimetic design, and by contributing to solutions to climate change, global health and food security challenges. As institutions, they have long been incubators for cutting-edge research in biology while simultaneously providing core infrastructure for research on present and future societal needs. Here we explore how the intersection between pressing issues in environmental and human health and rapid technological innovation have reinforced the relevance of museum collections. We do this by providing examples as food for thought for both the broader academic community and museum scientists on the evolving role of museums. We also identify challenges to the realization of the full potential of natural history collections and the Global Museum to science and society and discuss the critical need to grow these collections. We then focus on mapping and modelling of museum data (including place-based approaches and discovery), and explore the main projects, platforms and databases enabling this growth. Finally, we aim to improve relevant protocols for the long-term storage of specimens and tissues, ensuring proper connection with tomorrow’s technologies and hence further increasing the relevance of natural history museums.

Introduction

Natural history museums harbour the authoritative records of biological diversity across time and space and have always been meeting places for scientists, amateurs, and the public. By visiting a natural history museum and learning about nature, the lay citizen often tacitly endorses the information presented, so that museums are trusted resources, at a time when many other institutions are bitterly mistrusted (Foley, 2015). Natural history collections have a special role, in part because they not only serve public education, but in many cases they also are actively investigated to answer pressing problems in biology and beyond (e.g.,  Vane-Wright & Cranston, 1992).

Museum biological collections are more than meets the eye. Each specimen harbours many kinds of data, such as information on locality and collection parameters, or on associated pathogens, as well as biopolymers such as DNA and proteins, and metabolic compounds. This wealth of metadata across many specimens turns collections into powerful research tools, enabling scientists to test historic environmental hypotheses and carry out diverse studies ranging from public health & safety (as cornerstones in studies of environmental health and epidemiology; (Suarez & Tsutsui, 2004; Vo et al., 2011; DuBay & Fuldner, 2017); biomimetic design, where naturally-occurring architectures and systems inspire technological innovation (Jayaram & Full, 2016; Nirody et al., 2018); historical genomics (focussing on ancient alleles or past genotypes; (Bi et al., 2013; Besnard et al., 2014); to global and local change (tracking shifts in phenotype across specimens through time; (Jones & Daeler, 2018), something that a database of mere species observations cannot do. But natural history collections face challenges, many related to the need for long-term commitment to care and growth.

Survey methodology

This paper provides examples of how natural history museums have enabled discovery in evolutionary biology and environmental science; how that role has recently expanded to other fields of science; and how museums have helped foster the next generation of innovative thinkers.

Collections & types of museums

We broadly define natural history museums as institutions containing diverse physical specimens, including seed banks, substantial living, frozen, or dried tissue collections, and DNA banks, among others. These collections often include material from rapidly disappearing or extinct species, and often collected from the most inaccessible parts of the Earth. Collections may have innate, historic biases in taxonomic coverage and sampling design, which might need to be considered for their further development. Whereas historically natural history museums, in particular those in Europe, were linked to the colonialist enterprise, increasingly international specimen acquisition and study are conducted in strong partnership with local institutions. International agreements, such as the Nagoya protocol, rightly mandate such participation under the terms of Access and Benefit Sharing. Additionally, citizen science increasingly contributes to collections (see Box 1), which today are housed all over the world, and serve as gems of diverse global centres of cutting-edge research (see Fig. 1) and education, for an increasingly urban human population (e.g., Bates, 2007). Natural history museums may be located at universities without exhibits, or may include public exhibits, such as typically occurs in national, state or regional entities. In many cases, regional collections, and their exhibits, reflect and strengthen visibility, appreciation, identity and awareness of local culture, flora and fauna, therefore playing an important and confirmative role for the visiting public. Collections that span long periods of time reflect the history of science as well as changes in norms and values in society: what was sampled, how and why. This emphasis is especially visible in open, regional, collections on display. Regionality, therefore, can be considered a strength of collections by fulfilling an important role in sustaining knowledge and appreciation of the value of local biodiversity. Increasingly, regional museums were connected into the global network of collections.

BOX 1 Citizen Science examples.

Citizen science can contribute significantly to building collections, as for instance seen in many entomological collections that grow these days by amateur entomologists donating their well-curated collections for science. eBird and iNaturalist are excellent examples of connecting citizen with science in a highly-efficient manner (and then feeding into GBIF). The Gothenburg Natural History Museum (GNM) malacological collections have benefitted tremendously by citizen science efforts with devoted (‘advanced’) amateurs donating their often well-curated private collections. Based on these collections, which can be considered ‘environmental archives’, Bolotov et al. (2018) could infer from freshwater pearl mussel collections that morphology has changed in time correlated to environmental alteration and climate change. Based on historical and recent specimens from extensive geographical sampling, the authors concluded that climate change may well have accelerated the population decline in pearl mussels over the last 100 years. The study underlines the importance of preserving large collections (many individuals) to enable meaningful statistical analysis of morphological measurements. Below is another example from the GNM concerning garden slugs sent in by the general public.

Since 1986 the GNM has offered a slug-identification service to the public. The project was initiated as the invasive Spanish slug (Arion vulgaris Moquin-Tandon) began to spread rapidly over the country, prompting the need to establish a way to follow the spread and the colonisation process. As the slug spreads passively, mainly by the trade with ornamental plants and also with garden soil, it easily establishes in residential areas, where it mass-propagates and causes severe damage to vegetables and plants. A proper determination of these species requires dissection, a service offered by GNM. The project was advertised on TV, radio, in the newspapers and on museum web pages and GNM even offered to send out a transportation box which could easily be returned by mail. The response from the public was immense, and up to today GNM has received >6000 samples with slugs from all over the country. Together with the box, GNM sent out a questionnaire asking information about first year of occurrence, possible way of introduction etc. After determination of the specimens the senders got information of species identity, and in case A. vulgaris was concerned also advice for control measurements.

The project has been beneficial for both the gardeners and the museum. The colonisation process could be followed in detail and much information on the biology and behaviour of the species, as well as on the garden fauna of snails and slugs, was obtained. The latter included several other invasive species, from different parts of the country, and the development of this fauna over more than four decades could be monitored both geographically and chronologically. Furthermore, as at least one specimen from each species in the samples was preserved inEtOH, extensive material is available for DNA analysis, which has proved highly useful as the taxonomy of the Spanish slugs is complicated, involving hybridisation with other native and introduced species (Von Proschwitz, 1997).

Distribution, redundancy and digitization of collections

The distributed nature of the world’s museum collections increases long term data security. Collections of natural objects are vulnerable to theft, fire and water damage, but the distributed nature of collections and in some cases duplicate repository of specimens ensures persistence, such as for the tens of thousands of specimens of plants stored at the Berlin Herbarium that were almost completely destroyed during World War II. In contrast, the Butantã Museum in São Paulo had a world-renowned collection of 85,000 snakes and 500,000 arachnids that were not duplicated and shockingly destroyed by a fire in 2010 (Phillips, 2010). Despite these concrete recent examples of natural heritage loss, the infrastructure of many museums remains underfunded, exacerbating their vulnerability. A grim example is the Brazil National Museum in Rio de Janeiro, where a fire destroyed an estimated 90% of the collections in several divisions in September 2018 (Phillips, 2018), causing dramatic loss of both knowledge and data.

Creating overlap in collections, especially for extant species and genetic resource collections, is key to ensuring the longevity of these samples and associated data. Initiatives such as the Global Genome Biodiversity Network (GGBN; Droege et al., 2016) aim to collect, catalogue, and “democratise” genomic resources across global collections, covering 45,708 species (as of 30 September 2019). This enterprise represents an important step in the direction of a distributed collection, but would benefit from more coordination and financial support for data security, complementarity, and redundancy among collections. These aims are included in the mission of the pan-European Distributed System of Scientific Collections (http://www.dissco.eu) initiative, which aims at “Providing hard evidence on our planet’s natural diversity”, by enabling the transformation of a “fragmented landscape of the crucial natural science collections in Europe into an integrated knowledge base”. DiSSCo claims it “will allow Europe’s researchers and technology professionals to share and reuse the data linked to collections across disciplines and borders. It will mobilise and harmonise science collection data (collection metadata, traits, images, metabolites, nucleotide sequences, distribution or ecological information) and make them available as part of a highly connected linked-data graph”. In the US, the Integrated Digitized Biocollections (iDigBio) initiative supported by the National Science Foundation (NSF) has helped make data and images for millions of biological specimens available in electronic format “for the research community, government agencies, students, educators, and the general public” (Page et al., 2015). iDigBio serves as “the coordinating center for the national digitization effort” fostering partnerships, innovations, and content, but, crucially, is not assured of long-term funding by the NSF or other federal agencies. In contrast, DiSSCo, which has now been accepted into the EU’s Infrastructure Roadmap, started by securing government buy-in, with content being part of operational costs. Because it has become an established and recognized entity, governments fund the infrastructure because they need the services.

Figure 1 The centrality of natural history collections to evolutionary biology and public understanding.

Users, contributors and stakeholders of natural history collections are indicated; yellow arrows represent data flow, green arrows the flow of specimens.

Despite their immense value, natural history museums are facing grand challenges. Taxonomic expertise is decreasing for many organism groups or is not represented in the curation of some collections. Funding often relies on public sources and may be adversely affected by political and socio-economic changes, comprising the long-term continuity of a museum’s activities (see for instance Andreone et al., 2014). New international regulations on the collection, export and use of specimens for non-commercial and commercial purposes are now increasing administrative burdens and may prevent further development of collections. Cross-institutional, international coordination of secure data standards has not yet been fully realised. At the global level, the Global Biodiversity Information Facility (GBIF) is “an international network and research infrastructure funded by the world’s governments and aimed at providing anyone, anywhere, open access to data about all types of life on Earth”. As the main global biodiversity database, a large proportion of the >1 billion records comprise observations rather than specimens (see below). Smith, Johnston & Lücking (2016) and Yesson et al. (2007) discuss issues regarding data quality in GBIF, such as unreliable taxonomic identifications in the absence of voucher specimens, and non-global coverage of species distribution data. Verifying suspect occurrences through niche modelling, based on verified and geo- and DNA-referenced occurrences of the same species, is a step toward identifying unreliable records.

The value and diversity of biological specimens

All collections ultimately contain, or are dependent on, specimens. A specimen may consist of a complete organism (collected by naturalists over the past few centuries) or parts of a single individual organism. Increasingly, meta-data associated with the physical specimen, the ‘extended specimen’ (Webster, 2017), add value and increase data richness through videos, sound recordings, information on habitat, and photographs. For example, for birds, the extended specimen may be comprised of records of the song, or videos of behaviour of those organisms, prepared in a way that preserves them for the future. Bioacoustic tools provide unique collections that can include some of the last known evidence of extant species. Likewise, several films exist (e.g., https://www.youtube.com/watch?v=nAzqGn-LHCw) portraying the behaviour of animal species, such as the Tasmanian tiger Thylacinus cynocephalus, the golden toad Incilius periglenes, and the Hawaiian crow Corvus hawaiiensis, that are now extinct in the wild. Museums currently host increasingly diverse collections, which, in addition to DNA and tissue banks, may be generated by core genomic facilities or imaging labs (isotopic, X-ray computed tomography data [CT], scanning electron microscopy [SEM] images). Examples now include rich stores of high-resolution CT data generated from museum specimens, which allow investigators to look inside material in a largely non-destructive way. These require different storage resources from those that traditionally constitute museum infrastructures, namely large-scale and secure long-term storage of image data. Integration of different data streams will allow bridging among disciplines and the involvement of fields underrepresented in natural history museums, such as engineering, biomedical sciences, and art. For instance, biomimetic design can benefit strongly from inspiration from natural history collections (for examples from robotics see Jayaram & Full, 2016), or solutions to global health or food security challenges can be based on exploration of natural history specimen collections (see Box 2 Specimens and pathogens).

BOX 2 Specimens and Pathogens.

Museum collections have provided fundamental infrastructure for identifying and mitigating emerging pathogens and zoonotic diseases by public health agencies (Dunnum et al., 2017) such as the Centers for Disease Control (CDC). When a new pathogen emerged in 1993 in the southwestern United States, rapidly killing 7 people, authorities needed to know: had this pathogen accidently been released into the wild, or was it a newly evolved pathogen, or had the virus always been present and simply not previously identified? Without the availability of specimen archives, scientists would not have been able to efficiently determine the pathogen source and answer these fundamental questions. Large tissue archives from the Museum of Southwestern Biology and other museums (Yates et al., 2002) enabled virologists to quickly identify that this previously unknown zoonotic hantavirus pathogen was hosted by the locally common deer mouse (Peromyscus maniculatus). This virus is apparently transmitted to humans through inhalation of viral infected feces. Subsequent emergence of other human cases elsewhere in the Americas (e.g., Chile, Argentina, Panama), but with higher mortality, mobilized other specimen-based research efforts that identified other new strains of hantaviruses in many rodent species over the next 2 decades and on multiple continents. More recently, museum specimens of other groups of mammals were screened, leading to a radically reshaped understanding of hantavirus evolution, ecology and host occurrence (Yanagihara, Gu & Song, 2015). Not only were more rodent host species for these viruses identified, but numerous species of shrews, moles, and bat species worldwide also harbor their own hantaviruses. These specimens originated from multiple continents and the new discoveries significantly expand the potential risk to humanity of these pathogens and increase the burden on public health systems worldwide.

Other examples of pathogen outbreaks, such as the chytrid fungus in amphibians, have examined historical progression of diseases often decades or centuries after the outbreak (Schmitt et al., 2018). The impact of an invasive bacterial pathogen from poultry on native songbirds has been studied using avian tissue samples collected just prior to the outbreak in the eastern US, albeit without any foreknowledge of the impending epizootic (Hess, Wang & Edwards, 2007; Shultz et al., 2016). The Norwegian fish fauna is well documented in the scientific collection of the Natural History Museum, University of Oslo and collection material was screened for monogenean ectoparasitic flatworms of the genus Gyrodactylus that were (unintendedly) collected along with the fish (Zeyl et al., 2012). This yielded 13 flatworm species that are new to science, and an additional seven parasite species new to Norway. Three Gyrodactylus species were also recorded from new fish hosts, and in particular G. pterygialis appeared to have a broad range of host species, helping fishery biologists tremendously in understanding and managing fish populations. From plants, Yoshida et al. (2014) and Yoshida, Sasaki & Kamoun (2015) used potato herbarium in order to determine the genotype of the Phytophtera infestans strain that caused the great Irish potato famine in the 19th century (and concluded it was a ‘one-off’ type, never seen again). Herbarium DNA was also crucial in discovering ancient alleles in the grass Alopecurus myosuroides that are relevant to herbicide resistance but pre-dating human influence (Délye, Deulvot & Chauvel, 2013). Studies using genomic data of a 5,310-year-old maize cob (Ramos-Madrigal et al., 2016) have shown that our understanding of the process of domestication and early selection pressures needs adjusting.

Specimens are at the heart of the discovery process and technological advances are increasing the number and diversity of possible questions that can be addressed (e.g., Schmitt et al., 2018; see below). For instance, bone fragment identification using collagen barcoding was difficult to imagine before the rise of LC-MS and other technologies, but Welker et al. (2015) and Horn et al. (2019) used proteomics to identify Palaeolithic fragments of mammal bones and extinct species, respectively. Genomic analyses of single bone fragments can inform on the evolutionary and demographic history of our own species (e.g., Slon et al., 2018). Future technologies may include more advanced chemical, biochemical, isotope or micro-anatomical surveys, making specimens even more critical because they connect diverse data streams and facilitate data interoperability (see Box 3 for ‘best curation practices for the future’). At the same time, maintaining specimens is key to repeatability—a core requirement of science.

BOX 3 Specimens and best curation practices for the future.

Collecting. Recommendations for best preservation techniques for new specimens during field collection are as important as the final storage conditions for improving specimen long-term utility for genomics (Matos-Maraví et al., 2019). Documenting treatment practices is also key to facilitating future analyses enabled by as yet undiscovered technologies. The plethora of technological uses of museum specimens calls for a re-evaluation of how specimens are preserved. For centuries, plants have been pressed, animals mounted, marine specimens ethanol- or formalin-fixed and fungi dried. Although these standard preservation methods should still continue, if only because they constitute the bulk of biological collections thus far and have a proven track record of fostering discovery, whenever possible researchers should try to sample additional types of specimen parts, and organs and meta-data.

Storing. More studies need to be undertaken to improve relevant protocols for the long-term storage of specimens and tissues. Like digitization, banking of genetic resources by museums is an area of rapid innovation, particularly as next-generation sequencing methods have become more common. As museum tissue collections are accessed more frequently for genome projects, it has also become clear that the preservation standards and types of tissues preserved in museums are often inadequate for supporting the genomics enterprise. For example, a typical museum tissue sample from a bird, even if frozen in nitrogen hours after sacrifice in the field, yields DNA qualities and lengths unsuitable for 3rd generation long-read sequencing platforms such as PacBio and Oxford Nanopore. Such technologies rely on the use of long DNA fragments to start with, requiring specimen tissues be frozen immediately (within >10–15 min) upon collection. Although it may be difficult to use liquid nitrogen in the field, one solution is to use so called Dry Shippers, which are dewars designed for safe transportation of tissues at the same temperature as liquid nitrogen but without actually containing any free liquid nitrogen. Such shippers are routinely allowed for transportation back to the lab by airlines and can often hold cold temperatures for ∼3 weeks. Innovations in cryogenics are likely to transform collecting of genomic resources by museums in the future. But even here, some of us have noted poor DNA retrieval from tissues collected with standard field-protocols and ultimately preserved cryogenically (S Edwards, pers. obs., 2019). For birds, best practices for genome sequencing may not include freezing in the short term, which can fragment DNA, but rather unfrozen archiving of blood, which will preserve the longest DNA fragments. What seems to be most important is that DNA (and RNA) degradation is stopped as fast as possible after collection. For example, for birds, one way to achieve long DNA fragments for next-generation sequencing is to use unfrozen blood stored in Queen’s lysis buffer (Seutin, White & Boag, 1991), which has been used by ornithologists for decades and takes advantage of birds’ nucleated red blood cells. Blood stored in this way, with minimal shaking that will cause shearing, is a reliable source of high molecular weight DNA and has been shown to yield better genome assemblies than DNA retrieved from museum-grade frozen tissues (S. Edwards, G. Bravo, pers. obs). An alternative could be to store collections in the field in DMSO, although this appears to prevent RNA sequencing (Irestedt, unp. data). On the other hand, whether or not EDTA or 95% EtOH was used for DNA sample storage can be important too for successful long read sequencing (MI, pers. obs.). Still, we can take comfort that even from dried, centuries-old traditional specimens, valuable genetic information can readily be obtained by hybrid-capture and other approaches (Bi et al., 2013; Staats et al., 2013). Such best-practices for fieldwork and storage of genetic resources needs to be shared more widely and rapidly among the museum community. A useful platform for identifying both repositories and tissues for a wide range of taxa, often called biobanks, is provided by the Global Genomic Biodiversity Network mentioned above (GGBN, see http://www.ggbn.org/ggbn_portal/).

Below we outline various updates in storage of genetic resources for both animals and plants, highlighting issues facing museum curators and collection managers looking at the future.

Genomics is a key source of information and rapidly changing area in which the scope and potential of future applications are particularly promising. Nonetheless, there are several factors known to limit the utility of specimens for genomic analyses. For example, using ‘methylated spirit’ (methanol containing alcohol) instead of pure alcohol for field preservation of animal tissue can severely hamper retrieval of usable DNA later on (Post, Flook & Millest, 1993) (see Box 1). Heat treatment of plants, as typically applied in most historic herbarium collections, was found to lower genomic copy numbers but not cause significant miscoding lesions (Bakker, 2015; Staats et al., 2011). Conventional X-rays (as opposed to X-ray computed tomography with digital imaging) of mummies and bone or using pesticides on insect collections all negatively affect or destroy DNA (Gotherstrom, Fischer & Linden, 1995). Use of formalin to preserve specimens limits extraction of usable DNA from both animal or plant tissues as it causes cross-links among DNA molecules, preventing PCR (Ruane & Austin, 2017; McGuire et al., 2018).

For historic samples, significant progress in securing biopolymers has been made and museum and ancient genomics has attracted considerable interest, from researchers and industry (Hofreiter et al., 2015; Lindqvist & Rajora, 2019). Still, although some DNA sequencing technologies work well with degraded DNA, such as in herbarium DNA using Illumina sequencing (Staats et al., 2013; Bakker et al., 2016; Hart et al., 2016), single-molecule, ‘3rd generation’, genome sequencing will never be applicable for most museum-preserved specimens given the fragmented nature of their DNA. ‘Re-sequencing’, i.e., sequencing and mapping reads from multiple organisms against a related reference genome sequence, has been successful in museum plants, fungi and insects (Bi et al., 2013; Staats et al., 2013). For organisms with relatively small genome sizes, such as birds, the price for re-sequencing a genome from a study skin has become so low that curators of bird collections may consider to actually requiring complete genome sequencing for tissue from old museum samples. In this case, all parties ideally would benefit, the user for having access to the specimen, the museum for putting a halt to further specimen deterioration (as the genome sequence has been generated), and the next user for have both specimen and genome sequence available. That said, it is difficult to predict how DNA extraction techniques may evolve, and perhaps require considerably lower tissue amounts to produce higher DNA yields, meaning that high-throughput DNA extraction without an immediate use (DNA banking) is not an obvious choice for museums.

Often scientists endeavour to see inside museum specimens. Previous approaches such as dissection or histology are invasive techniques that necessarily result in the destruction of other data. Although recent imaging techniques (diceCT; Gignac et al., 2016) enable largely non-destructive work on these questions in non-model organisms preserved in alcohol, they do not completely ameliorate data loss due to selectivity in field materials preserved. Specimen field preparations may include freezing fresh tissue for DNA, preserving skeletons and skins but removing most internal organs and muscles. The latter obviously limits the kinds and diversity of research that can ultimately be performed on such specimens. For example, the vocal organ of birds was often not collected in birds despite the perceived importance of bird song and other vocalization. Some now broadly used imaging techniques (e.g., diceCT) have not been studied for their effects on DNA/RNA amplification from formalin or alcohol-preserved specimens, and it is unknown if they further inhibit downstream molecular work involving these specimens.

For most large multicellular organisms, it is challenging to collect large numbers of tissues. However, more portions of an organism can be feasibly preserved before discarding tissues when making new collections, particularly of common, easily accessed species. For example, at the Museum of Comparative Zoology at Harvard University, a typical avian specimen is now accompanied by 7–10 cryovials filled with DNA- and RNA-ready tissues from different organs, as well as at least one tube of unfrozen but refrigerated blood for genome sequencing. Such sampling will no doubt pose space challenges for long-term storage (which could be partially solved through the use of space-efficient biobanks), but is essential for, for instance, a deep understanding of the effects of anthropogenic change on biodiversity (Schmitt et al., 2018). Integrating new imaging techniques into museum work flows will increase documentation prior to destruction (e.g., for genomic work). For instance, the Thermal Age Web Tool (http://thermal-age.eu/) was developed to help collections managers and users to quantify the risks associated with destructive analysis of specimens, based on calculated ‘thermal ages’ (Smith et al., 2003). The Synthesis of Systematic Resources programme (see http://www.synthesys.info/joint-research-activities/) provides further recommendations for non-destructive sampling of museum specimens and decision analysis as to how to best sample specimens for genomic research. Lower price points for acquisition of genome data and some imagining techniques makes defining these best practices more urgent.

Transcriptomics and Epigenomics. The ever-increasing number of genome sequences becoming available can be efficiently explored in terms of gene function through transcriptomics - the sequencing of all transcribed mRNA expressed at a certain time, physiological or developmental state for a particular tissue. In this way, the 1Kite (http://www.1kite.org/) and 1 kp (https://sites.google.com/a/ualberta.ca/onekp/) projects, assembling 1,000 transcriptomes of insects and plants, respectively, have expedited progress in both comparative and functional genomics and a better understanding of gene function across these clades (see for instance Gitzendanner et al., 2018). We would expect future specimens to play an increasing role in this respect, but only if we make sure to store our specimens in such a way that RNA is preserved, for instance by rapid cryogenic storage of use of RNA-friendly buffers like RNA-later. Additionally, a diversity of epigenomics approaches, such as methylation, Chip-seq and ATAC-seq, are emerging and potentially of great use to the field of evolutionary biology (Grayson et al., 2017). Epigenomics is already commonly applied in evolutionary studies of adaptation and development, and has recently made headway in examining epigenetic maps of extinct human and plant populations (Llamas et al., 2012; Gokhman et al., 2014; Smith et al., 2014; Smith, Monroe & Bolnick, 2015; Sackton et al., 2019). Best practices for preservation of biomaterials for epigenetics has not yet been discussed in the literature, and will be an important additional consideration for museum curators in the future.

Proteomics. Given future technological developments, it is likely that proteomes will be determined and used for further functional studies across the Tree of Life. Additionally, collagen from bone tissues have been demonstrated to give species-level amino acid variation from specimens several millions of years old using a ZooMS approach (Welker et al., 2015). Portugal et al. (2010) report on proteomics in museum egg specimens and conclude that current approaches to proteomics in such specimens may be limited in coverage of the proteome. In any case, storing tissues in the best possible ways for proteomics, ideally, cryogenically in order to stop proteases from working, now ensures that such analyses can be conducted in the near future.

Secondary metabolites. Compounds such as alkaloids, glucosinolates, furanocoumarins, flavonoids or terpenes can be measured in museums tissues, especially from plants (Berenbaum & Zan, 1998; Colegate et al., 2014; Mithen, Bennett & Marquez, 2010; Raffauf & Von Reis Altschul, 1968). Access to such metabolites enables testing historic biological hypotheses such as past responses to change in herbivores and climate; but also in case of invasive species and testing what secondary compounds may have driven invasive success in species such as Vincetoxicum nigrum (Asclepiadaceae) (Liede-Schumann et al., 2016).

Stable isotopes. Advancement of techniques for studying specimens include measuring of stable isotopes, allowing monitoring environmental/atmospheric changes over time, given a time series of museum specimens (reviewed in Schmitt et al., 2018). Because elements are not expected to degrade over time like biopolymers do, proper specimen storage conditions for isotope analysis is probably not critical. Limiting factors in such studies now is the availability of robust spatial sampling and time series of specimens for analysis. Properly tracking the vast quantities of data that are generated in these analyses directly to the specimens is also a challenge (Pauli et al., 2017).

Non-standard tissues. Classical botanical specimens comprise branches with leaves and fertile organs (flowers, fruits). For some vertebrates, such as birds and mammals, it is primarily the external morphology that is preserved in collections, whereas for amphibians, reptiles and fish, preservation in formalin and/or alcohol can yield DNA sequences with some effort (Ruane & Austin, 2017; McGuire et al., 2018). Many biobanks, particularly in US museums, now also include samples of frozen blood and tissue from vertebrates, typically heart, liver and muscle. However, many other parts of organisms not conventionally stored in museums are becoming important in the effort to monitor global change. For instance, there is great interest among diverse scientists in investigating the microbiome of species—including the bacteria present in the digestive system, and what roles they may have to the species’ adaptations to the local environments (Roggenbuck-Wedemayer et al., 2014; Alivisatos et al., 2015; Ingala et al., 2018). Similarly, tree bark may provide important information on chemical defenses of plants and hold implications for medical applications (Maldonado et al., 2016). Transcriptome studies in vertebrates are becoming increasingly common as a means to study species’ ability to adapt to changing environments and anthropogenic change (e.g., Zhang et al., 2014) and are widely used in phylogenomics of invertebrates and plants (Wen et al., 2015). Such studies encourage careful sampling and preservation of whole organisms or all organs separately when appropriate.

Specimen collections enable answers to a large number of other scientific questions, some of which have not yet been posed. The earliest museums facilitated interactions among scholar-travellers, to share observational data from across the planet and to help build the core of what would become natural history and modern evolutionary biology. Increasingly, museums are leveraging new data from their specimens, and this integration of data types allows training in techniques that bridge among disciplines, as well as the generation of data sets that are of relevance to disparate traditional fields such as engineering, biomedical sciences, and art. Today natural history museums serve increasingly as a nexus for work that disregards disciplinary boundaries and addresses questions we did not know to ask before. Because collections provide the opportunity to rigorously examine diverse aspects of taxonomic, morphological, genetic, and chemical variation across vast temporal and spatial scales, they can help diverse scientists bridge the gaps between traditionally distinct disciplines. Museum spaces ideally are filled with students who learn to think anti-disciplinarily and appreciate the importance of the specimen. These spaces can therefore be considered ‘Innovation Incubators’ where a next generation of critical thinkers in biology and beyond will be trained, embedded within the prerequisite of traditional taxonomic museum expertise.

A specimen constitutes a voucher, not only of the actual individual sampled at the time, but often also of its locality–including information about the soil and other biotic and abiotic conditions in which the specimen was collected (see below, the ‘holistic specimen’). In fact, the specimen can be seen as the outcome of a combination of genotype and past environmental change or conditions (e.g., Holmes et al., 2016), and a well-curated collection captures the variation in phenotype as well as genotype (see Bi et al., 2013; Rowe et al., 2011; Staats et al., 2013; Ruane & Austin, 2017). For instance, Cridland et al. (2018), comparing SNP patterns from historic museum and living specimens of bees, could not only infer ‘rapid change’ in genetic composition of honey bees in California, but also identify historic genotypes in candidate genes possibly involved in adaptation to new niches. Another, non-domestic, example involved reconstructing the shift to C4 photosynthesis in grasses using DNA from a 100-year-old Malagasy herbarium specimen for which both phylogenetic placement and genetic regulation of C4 photosynthesis could be assessed (Besnard et al., 2014). Therefore, specimen collections can provide a powerful reference for functional genomics studies, in a world where assessing phenotypes of different genotypes, retrievable from the specimen, is essential.

Specimens vs. observations in digital collections

Troudet et al. (2018) describe how over the past decades the proportion of specimen-based occurrences in GBIF has decreased from 68 to 18%, in favour of observation-based occurrences, either expert-validated or not, mostly from contributions by citizen science efforts such as iNaturalist and eBird (see Table 1, and Box 1). This will have affected repeatability and ‘richness’ of systematics and evolutionary studies and the authors urge that “when impossible to secure, voucher specimens can be replaced by observation-based occurrences”, particularly when combined with ‘ancillary’ data such as recordings, pictures, DNA samples etc. In cases where ethical, conservational, or practical concerns exist, observation data instead of collected specimens provide additional (or occasionally substitute) contributions to our knowledge on where and when particular species occur. Recorded sightings, such as those from iNaturalist or e-Bird, include occurrences of diverse temporal range, and are pretty much the only observation-based data that are allowed in GBIF. In addition to such observations recorded in the field, however, collected specimens, when available, offer additional options for confirming or extending the original work using new analytical techniques. Similarly, sound recordings can be re-studied within the context of new evidence, leading to reciprocal illumination. Boakes et al (2010) asked how species distribution data for a well-known clade (Gallus) from museum collections compared across literature records, banding (ringing) data, ornithological atlases, and birdwatchers’ trip reports in terms of completeness and consistency. They found that “..primary sources of biodiversity information are subject to a range of biases that fundamentally affect their interpretation and therefore their reliability in measuring biodiversity change” (Boakes et al, 2010).

Table 1 Major global and local aggregators of biodiversity data.

Acronym	Mission	Funding; scope	Type of data	Volume of records (M)	
ADBC	Advancing Digitization of Biodiversity Collections	US			
ALA	Atlas of Living Australia. https://www.ala.org.au/	Australia	Observations, specimens	84.8	
BOLD	Barcode of Life Database	Canada; global		7.76	
DiSSCo	Distributed System of Scientific Collections; digitization and databasing of european specimen collections	Europe	Specimens	1500	
eBird	Citizen science: the world largest biodiversity-related citizen science project, gathering information on bird sightings, archive it, and “freely share it to power new data-driven approaches to science, conservation and education.” https://ebird.org/home	Global	Observations	100 ‘yearly’	
EOL	Encyclopedia of Life; Global access to knowledge about life on Earth	Australia, Egypt, US; global	Species descriptions	<1,9	
GBIF	Global Biodiversity Information Facility	Global	Observations, specimens	1300	
GGBN	Global Genome Biodiversity Network	Global	DNA, tissues, environmental samples	3.8	
GloBI	Global Biotic Interactions; species interaction data by combining existing open datasets. https://www.globalbioticinteractions.org/	US; global	Species interaction data e.g., predator–prey, pollinator-plant, pathogen-host, parasite-host	>0.7	
HOLOS	Berkeley Ecoinformatics Engine: accessing and visualizing integrated biological and environmental datasets to address questions of global change biology. https://holos.berkeley.edu/	US; global	Different kinds of biological and environmental datasets	n.a.	
iBOL	International Barcode of Life; extending BOLD’s coverage. iBOL’s forthcoming BIOSCAN will activate a biomonitoring system for half the world’s ecoregions, metabarcoding assemblages and studying species interactions from 2,500 sites. https://iBOL.org	Canada; global	DNA barcodes and metadata	see BOLD	
IDEA	Island Digital Ecosystem Avatar; place-based systems ecology for building simulations of social-ecological systems	US; Moorea	Specimens, observations	?	
iDigBio	Integrated Digitized Biocollections; digitisation and databasing of US specimen collections	US	Specimens	117.5	
iNaturalist	Citizen science: one of the world’s most popular nature apps, sharing observations globally; https://www.inaturalist.org/	US; global	Observations	<1	
LifeWatch	Biodiversity research, -management and -conservation priority setting	Europe	Research tools	n.a.	
iSpot	Citizen science: experts helping citizen community to identify its wildlife observations. https://www.ispotnature.org/	UK; global	Species identifications	0.030	
MoL	Map of Life; providing species range and dynamics information and species lists for any geographic area. https://mol.org/	Global	Occurrences, observations	8.8	
NCBI	National Center for Biotechnology Information	Global	Nucleotide and amino acid sequences; genome annotations	0.37 species covered	
NEON	National Ecological Observatory Network; continental-scale environmental data, infrastructure for research, educational tools to work with large data. https://www.neonscience.org/	US	Observations	?	
OToL	Open Tree of Life; construct a comprehensive, dynamic and digitally-available tree of life by synthesizing published phylogenetic trees along with taxonomic data. https://tree.opentreeoflife.org/	US; global	Phylogenetic trees and taxonomies	2.6 OTU’s in taxonomy	
Traitbase	Ecological species characteristics, individual level species information. https://traitbase.info/whatis	Spain; global	Specific characteristics e.g., body size, diet or fecundity	?	

The need for continued and comprehensive collecting

Continued field collecting ensures that museum specimens and data will be accessible for future generations and secures future access to time series of specimens, collected continuously over decades or even hundreds of years. These long-term archives provide valuable and unique information (Graham et al., 2004) on changes in species composition in environments and habitats, due to factors such as climate change, human-mediated nitrogen deposition, or other anthropogenic activities (Meineke et al., 2019; Meineke & Davies, 2019). An example is a large survey and collection of marine invertebrates from the Swedish west coast from the 1920’s and 30’s conducted by the Gothenburg Natural History museum, in which the exact sample locations could be deduced, and consequently revisited during a new survey in the 2000’s, revealing a 60% loss of biodiversity (Obst et al., 2017). Indeed, specimens collected by researchers 200 years in the past can be compared with contemporary (and future) sampling—as long as these collections and institutions persist.

Whereas the value of field collecting may be obvious within the broader museum community, justification for collecting specimens is not always clear for society and even some scientists (e.g., Rocha et al., 2014). There can be additional challenges for collectors in Buddhist countries where killing living things is problematic because of prevailing philosophies. In some sectors of the US, it is clear that the public misunderstands the mission of museums and does not appreciate the need for continued responsible collecting. One recent example is the unwarranted overreaction against scientific collecting of a bird specimen from the Solomon Islands, information about which was placed on the web by well-meaning media directors at the American Museum of Natural History (Johnson, 2018). This sad event, which resulted in death threats and cyber-harassment of the scientist involved, shows that many people see only the destructive aspects of collecting of individual specimens, but do not connect this act with the many beneficial services of museums to science and society. Additionally, some laypeople did not appreciate the relative insignificance of scientific collecting as an agent of species loss as compared with habitat loss and introduced or feral predators, such as house cats (see Marra & Santella, 2016; Woinarski, Legge & Dickman, 2019). In this case, the public seemed to place undue emphasis on the loss of life incurred by collection of a single bird, suggesting much more relevance of an emotional response rather than a scientific appraisal of the true impact of collecting data on populations.

Amending the idiosyncracies and biases of natural history collections

Historically, some of the largest collections have grown through the acquisition of specimens through professional collections, such as the ornithology collections at the Museum of Comparative Zoology at Harvard. In the past 50 years, natural history collections have instead grown more through the research programs of individual scientists. Collections growth by these means can result in spectacular coverage of specific clades or localities, but often results in patchy coverage of biodiversity as a whole, or coverage in a way that is not maximally useful for studying the effects of global change on organismal diversity. Collections have also tended to emphasizing rarities (i.e., occurring as single individuals per species), a biological phenomenon which may actually be commonly-occurring, for instance in tropical herbivorous insects (Novotný & Basset, 2000). Such biases often result from efforts to conduct general collecting (Ter Steege et al., 2011). Like research-driven collecting, these efforts at generality have resulted in an invaluable reference specimen base in today’s museum collections, allowing comparison with living specimens, identifying relatives of medically- and economically-important species (for instance melon, Sebastian et al., 2010), or testing historic biological hypotheses (e.g., Délye, Deulvot & Chauvel, 2013). On the other hand, this patchy tradition of biological collecting has come at a cost to easily comparing organisms across large geographic regions or across temporal spans. For example, evolutionary biology would benefit from being able to analyze more common species represented in collections worldwide, because this would allow assessing phenotypic variation at much broader scales. In addition, assumptions about species ID based on morphology may be falsified by DNA data DeSalle, Egan & Siddall, 2005—but also the reverse—revealing an unexpectedly high level of cryptic diversity in certain groups (e.g., Hebert et al., 2004).

Increasingly, growth of museum collections is the result of their increased relevance for ecological studies, in addition to input from taxonomically-focused collecting activities, linked to specific inventories and research projects. Whereas museum staff and associated researchers and students still undertake expeditions to increase collections and make them available for future generations, some collections also now come from large scale ecological studies (e.g., NEON in the US), although they are often not conducted and archived in ways that maximize specimen value (Cook et al., 2016a; Cook et al., 2016b). Still, such systematic collecting raises the issue of what mandates should guide biological collections growth in the future. One such mandate would require a concerted effort of museums globally to collect and archive specimens in a coordinated manner that would help document current biodiversity and variation of common species across the globe. Such an effort was originally planned to be conducted by NEON in the United States, but in some cases has fallen short of this goal (Cook et al., 2016a; Cook et al., 2016b). Other ventures include the above-mentioned BOLD (with iBOL extending its coverage) which holds 7.76M DNA barcode records across 0.31M species, many of which are commonly-occurring and for a large part with vouchers present/documented. Future collections should continue to expand with specimens sampled widely across biodiversity, but in addition should amass commonly-occurring species, which can serve as environmental monitors, especially when sufficient metadata is also collected.

Place-based discovery: different specimen data sets connecting to a location

When collecting efforts from the perspective of different clades have focussed on a particular location, an additional layer of comparative information emerges, that of species-correlations, environment and edaphic correlations, and the evolution of communities i.e., the full-suite of local biodiversity. Taking ‘place’ into account enables asking different kinds of questions and can lead to ‘place-based discovery’. As Miller (2007) puts it, “Places are not simply a semantic convenience. They are a meaningful lens for viewing the world because it is orderly with respect to geographic space”. As such, ‘place-based’ approaches in general focus on the characteristics and meaning of particular places as a fundamental starting point for a particular development or project. Especially in charity and community development work, place-based approaches aim at “giving power to the community in guiding systemic change” and therefore “being recognised as critical to responding effectively to certain community challenges” (http://www.qcoss.org.au/). In education, place-based approaches are thought to “identify, recover and increase the values of local cultural specificities” (Monardo, 2019). Analytically, place-based approaches were successfully applied in a study on community-membership predictions (Caughlin et al., 2013). In marine resources management place-based approaches, including human activities and use along with biodiversity conservation have proven important since long ago (Carollo et al., 2009).

Place-based learning and education is well developed (Gruenewald & Smith, 2014) and provides a context for local understanding and societal change. Natural history museums are well suited for hosting place-based activities, as well as making direct links between collections and associated data and societal activities and needs. The developing Island Digital Ecosystem Avatar (IDEA) project is one example (Davies et al., 2016), entailing “a systems ecology open science initiative to conduct the basic scientific research needed to build use-oriented simulations (avatars) of entire social-ecological systems”. Many specimens will have been collected and stored, for instance for DNA barcode reference libraries, for making this possible.

For biological collections and their associated and ensuing process of discovery, the place-based approach is relevant as it seeks to understand how local information and processes are interconnected with those at a larger scale. Best practices for biological collections include a precise geographical reference for each item, as is included in the Darwin Core (see https://dwc.tdwg.org/). When collected together, sets of items are necessarily place-based. In addition to standardised metadata directly associated with biological items (Kissling et al., 2018), many other types of information are place-based and can be collected at the same location and super-imposed on point specimen data. Examples include information about geology, ground and atmospheric chemistry, and archaeology. These, and other data layers—such as from GBIF (species occurrences), NCBI (DNA and amino acid sequences), Open Tree of Life (phylogenetic trees), Map of Life (abundance data), TraitBase (traits), GloBI (biotic interactions), see Table 1—can be associated or combined with geographical location through a geographical information system (HOLOS), integrating across diverse data types and enabling testing hypotheses concerning causal impacts—the ‘holistic specimen’. In a sense, this approach is comparable to correlative species distribution modelling (SDM) approaches such as using Maximum Entropy (MaxEnt; Phillips & Dudik, 2008), focussing on mostly abiotic and edaphic correlates.

In addition, the place-based approach can provide a baseline for understanding changes over time (Billick et al., 2013; National Research Council, 2014). An understanding of historical processes provides a means for predicting, or forecasting, how biological systems may respond to change in the near future. For instance, Willis et al. (2008) studied how climate change may affect phenology in angiosperm species in Concord, Massachusetts. Slingsby et al. (2017) studied the interaction between fire and climate change on species diversity in the South African Cape Floristic Region, allowing modelling of future vegetation response.

In general, the additional value of place-based approach for scientific discovery includes the serendipity of collecting data over periods of change identified later, as well as the interaction of researchers sharing an interest in the same geographical location or region (Michener et al., 2009). Place-based initiatives associated with larger networks (see Table 1) can provide access and understanding to a diversity of communities, which is both democratic and allows broad participation in discovery. Examples of such initiatives include developing new natural history museums (Darwin Initiative, www.darwininitiative.org.uk) and establishing a reference DNA barcode library (with vouchers for each specimen) of local biodiversity (Van de Bank et al., 2008; Janzen & Hallwachs, 2016) representing a place- (not clade-)based, regional, effort. In addition to natural history museums, the benefits of a place-based approach are also shared with field stations, botanical gardens, and biological reserves (National Research Council, 2014).

The Global Museum

By ‘Global Museum’, we mean the potential global community of natural history museum collections that increasingly are becoming interconnected through digitization, thereby enabling both synergy and increased power of resolution in testing (mostly) evolutionary biological hypotheses. With the Global Museum, scientists worldwide are able to study and sample each other’s specimens (including from remote collections), perform geometric morphometric analyses of shapes or generate high-resolution CT data and compare related images. Phenomic and extended specimen; Box 3 data on particular populations, species or actual specimens would be available at a large scale, from databases either as part of or outside the Global Museum, enabling correlation and modelling studies at a global scale, and testing evolutionary hypotheses with unprecedented power. As indicated above, many museums serve regional communities, and collections in such institutions usually reflect regional interests, fauna and flora, funding and research questions. Given that science is an international endeavour, the question can be asked as to whether evolutionary biology would be better served by enhanced ability to document and analyse patterns across regions, such as with the use of GBIF. For instance, for taxonomy, having the virtual, global, workbench of the Barcode Of Life Database BOLD (http://www.boldsystems.org; Ratnasingham & Hebert, 2007) theoretically allows taxonomists globally to harmonise species delineations by collectively analyzing and interpreting DNA barcode (and associated distributional) patterns from global rather than regional data sets. Boufahja et al. (2015) present another interesting example of connecting regional and global scale in marine nematodes.

Museum communities are increasingly not confined by a single, local physical space but able to distribute their reach through innovations in technology. Databases and other online tools enable international access and an array of novel platforms facilitate participation of a broad swath of the public in discovery and documentation, from undergraduate classes to young children participating in aspects of the scientific process. Examples of such activities include encouraging children to make observations of butterflies in drawers, thereby likely building their sense of biodiversity. Another example is citizen science projects in which volunteers help in interpreting and digitizing information on old collection labels, as has been done for the Paris Herbarium (http://lesherbonautes.mnhn.fr/) and for brachiopod fossils at the Swedish Museum of Natural History (see Box 1).

From a Global Museum perspective, we may ask whether phenomena such as global change have been effectively documented in collections in the past so that we can use the ensemble of past collections to forecast future conditions. For instance, collections can help scientists document how C4 photosynthesizing plants have spread during recent decades as a response to the global increase in atmospheric CO2 concentration (Besnard et al., 2014), or how species extinctions may be overrepresented in particular clades or areas (e.g., Ricklefs, 2006). Such work would be impossible without having the integrated, properly digitised and databased platform that a Global Museum provides.

Large international data sharing initiatives (e.g., LifeWatch ERIC, GBIF, Encyclopedia of Life, BOLD and iBOL, see Table 1) allow access to collections by scientists and the public living far from privileged historic western centres for inquiry. For instance, GBIF alone provides access to now over a billion records of specimens and observations from around the world. iDigBio, GBIF, and the Atlas of Living Australia (ALA) and its affiliated atlases are the world’s largest and best-developed biodiversity data aggregators and mobilizers. As outlined above, DiSSCo is a developing initiative of major significance that will unify natural science collections in Europe. With increasingly distributed access to large datasets and online portals to large-scale computational resources, analysis of the “big data” of biodiversity records can also go global (see Antonelli et al., 2015 for an example in global angiosperm biogeography and speciation). Digitization of collections will be increasingly important in this respect; there are many valuable but undigitized collections (and literature resources) residing in museums, not in the least outside of developed countries.

But a Global Museum is more than increasingly inter-connected biodiversity databases across the globe. In order to be effective and truly innovative it should be synergistic in answering scientific questions, in a way that single museum collections cannot do. For instance, linking data on phenomes, including genomes, proteomes, parasites, secondary metabolite chemistry, distributions, morphology and generally all ‘extended specimen’ metadata, across global museum specimen collections would enable testing new hypotheses and correlations, and with synergy not unlike that observed in studies using synthesis radio telescopes (Levanda & Leshem, 2010). Exciting examples of such studies can already be found for instance in Lamichhaney et al. (2019) where the authors focussed on phenotypical convergence and its possible genomic correlations.

For a synergistic Global Museum to happen across a wide spectrum of museum situations and funding sources, there would need to be ample coordination in terms of setting-up an operational infrastructure, governance structure and data standards. Issues such as data ownership visibility and transparency could perhaps be improved using emerging blockchain technologies (see below). The level of funding required for such an initiative, and what balance between national and global funding is preferred and over what time frame, are questions that need to be explored, probably by museum communities and federal funding agencies. Additionally, how will policy makers as well as the general public be convinced of the importance of a Global Museum as a top-priority? Comparisons are often made with astrophysicists and how they are able, as a united community, to leverage unprecedented large sums of funding for ever more ‘extremely’ and ‘overwhelmingly’ large instrumentations (see https://en.wikipedia.org/wiki/Overwhelmingly_Large_Telescope). How can global biodiversity scientists and museums achieve comparable goals? What role can existing platforms and museums play already? The best roadmap to follow towards a Global Museum is beyond the scope of this paper, but we do appreciate the reality of selling the concept to ‘the public’ may be a formidable challenge.

Increasing the relevance of museums through digitization and imaging

As indicated above, digitization of collections is important for achieving a global perspective on (big) biodiversity data. To facilitate the coordination of collection and databasing efforts between museums—a necessity to achieve a Global Museum—it is vital to increase awareness of what knowledge is available, not only at regional museums but across museums globally. This is most easily achievable through digitization of the collections. Moreover, such digitization also opens up the collections for a number of additional researchers interested in overall temporal or spatial patterns in biodiversity. iDigBio (see above) provides a good example of how digitization can be successful and provides outreach to a global user base. In its first 10 years, iDigBio and NSF have appropriately prioritized digitization of specimens that can drive collaborative research and answer specific biological questions. However, this specialization necessarily results in only a small fraction of available specimens being digitized.

For any information system to be successful, proper nomenclature and taxonomy of specimens involved are paramount, the bedrock of success. Names applied to specimens will change as science progresses but data will remain. Ideally, names applied to specimens are up-to-date but good museums do indeed worry about synonymy. A prime example of combining data bases and ‘vetting’ them is VertNet (Constable et al., 2010) which combines four global vertebrate networks: Mammal Networked Information System (MaNIS), Ornithological Information System (ORNIS), HerpNET and FishNet 2. These networks collectively mobilize over 52 million records from more than 70 institutions, representing about 70% of all the vertebrate species occurrence data that are accessible through GBIF. A good example for keeping track of synonyms is the Swedish Taxonomic Database DynTaxa (https://www.dyntaxa.se/), and for combining Biodiversity data bases see the Swedish LifeWatch Analysis Portal (https://www.analysisportal.se/).

A major question for the future is how the community should greatly expand the scope of digitized specimens. In the US, the Biodiversity Collections Network (BCoN) developed a strategy for the next decade to “maximize the value of our collections resource for research and education” envisaging building a network of Extended Specimen data (Biodiversity Collections Network, 2018; Webster, 2017; see also Biodiversity Collections Network, 2019). Just as incidentally collected historical specimens often prove useful for research questions not envisioned during the collecting event, it is also likely that specimens digitized without a specific research question in mind will prove useful for answering scientific or societal questions, especially if digitized on a large scale. But digitizing ‘blindly’ must of course be balanced with the pressure of meagre resources; enabling citizen scientists to assist offers a good solution (Rouhan et al., 2016). Large amounts of metadata remain to be digitized and would generate knowledge on biogeography (geographic data of specimens), disease spread (genetic material from parasites), biological interactions (pollination data), phenology, or shifts in species distributions (Suarez & Tsutsui, 2004; James et al., 2018). New advances in image recognition through deep learning using neural networks are also likely to enable easy identification of many species, such as already implemented in the iNaturalist platform, and hence help digitization.

Tracking specimen taxonomy

For research on temporal patterns of global change, such as global warming studies or analyses of movement of hybrid zones, museum records provide a unique source of historical records. Because they are backed up by physical specimens, records can be identified to individual species irrespective of how many taxonomic changes have occurred since specimen collection, an endeavour that is often impossible if the records were purely observational. However, an important caveat of this effort is that it is rarely obvious from online databases on what taxonomical opinion each identification was based. Indeed, to capitalize on the inherent strength of specimen-based records a citation to the taxonomic revision behind each identification would be ideal, but is not always available. The above-mentioned Biodiversity Collections Network (2018) report suggests exploring blockchain technology (https://en.wikipedia.org/wiki/Blockchain; Iansiti & Lakhani, 2017), which is a decentralized approach to data exchange that has already gained significant traction in the corporate, academic, and federal arenas and is applied to track items through complex supply chains. Blockchain technology involves an open, distributed ledger, or list of cryptographically linked records (‘blocks’), that can record transactions between two parties efficiently and in a verifiable and permanent way (Iansiti & Lakhani, 2017). It is considered to enable decentralised ‘incorruptible repositories of quality graded data’ (Hilsberg et al., 2018) and Biodiversity Collections Network (2018) expects it may provide “viable solutions for how data and products can be stored, linked and shared with collaborators and stakeholders, facilitating transparency and traceability. In addition, it is expected that it could be valuable for low-cost data-preservation (while enhancing data-reachability) as well as enhancing collection management (see also Hilsberg et al., 2018).

Combining specimen databases

Another challenge related to digitization and to the value of the physical specimens is that specimens may be one of the only clear, if frequently underutilized, ways to identify duplicates between different databases. For example, this challenge is often encountered among mammalian fossils where several databases, including ‘New and Old World Mammals’ (NOW) and ‘PaleoDB’, have been started independently. Together these databases completely document diversity for some groups, and out of ∼1,586 accepted species of carnivores and relatives (Carnivora, Hyanodonta and Oxyaenidae) in the fossil record, 1,460 species are included in at least one of them (Faurby, Werdelin & Antonelli, 2019). However, individually they are each highly incomplete (1,121 species, 6,385 records in NOW, 1,040 species (6,756 records) in PaleoDB). Analyses attempting to summarize the complete fossil record therefore needs to combine databases. For many uses, such as when http://onlinelibrary.wiley.com/doi/10.1111/2041-210X.12263/abstract speciation or extinction rates (Silvestro, Salamin & Schnitzler, 2014), it would be highly desirable to resolve redundancy between databases, a challenging effort that could be made trivial if both databases contained the museum specimen ids for the records.

Big clades, large collections

Digitization will be extremely challenging for the most diverse taxonomic groups, such as typically large collections of insects, although there have been attempts to automate digitization of such groups (Hudson et al., 2015). Generally, though, a lack of professional staff, infrastructure and facile technology are usually limiting this process. For example, a competition sponsored by NSF with US $1M (see https://www.nsf.gov/news/news_summ.jsp?cntn_id=133377) as an award for developing technology to automate the digitization of insect specimens and metadata (often from hand-written specimen tags) failed to identify a winner because the proposals from all competing teams fell short of award criteria. For any specimen collection, knowing its limits, in geographical and temporal distribution and in size, especially for the most common groups, may suffice for further scientific analysis (see https://www.analysisportal.se). Finer scale distributional data in such cases can easily be supplemented by citizen science initiatives (e.g iNaturalist), especially if associated with pictures or movies with smartphones that contain georeferenced and timestamped records that facilitate re-validation through inspection of the images by experts (see for instance iSpot; Silvertown et al., 2015 and Beespotter, https://beespotter.org).

Museums still harbour large amounts of undiscovered and undocumented information. The total number of specimens deposited in museum collections around the world may be as large as 1–2 billion (Ariño, 2010), and for herbaria an estimated 350 M specimens are known to be deposited in 3,400 collections world-wide (Soltis, 2017). Moreover, statistical approaches to estimate the size of collections agreed in 2010 that less than 5% of the universe of natural history collections data is available in databases such as GBIF (Ariño, 2010), although this fraction has been decreasing, with the fast increase of observation data in GBIF (see above and Troudet et al., 2018). Wilson (2003) noted that the smaller the organism the more poorly known the group to which it belongs, exemplified by fungi, nematodes and microbes. For instance, a random selection of specimens collected in a tropical rain forest and deposited in jars at a natural history collection resulted in the description of almost 200 new species of ichneumonid parasitoid wasps to science (Veijalainen et al., 2012). Bebber et al. (2010) described a comparable case for angiosperm species, with an estimated 35,000 undescribed species already residing in herbarium collections. An abundance of undescribed species is only the tip of the iceberg on the amount of data undiscovered and undocumented in the world’s museums. Again, enough professional staff and infrastructure is needed to tackle this problem.

In terms of species diversity, and in addition to the hard curation work by thousands of taxonomists, DNA barcoding reference libraries such as BOLD (Ratnasingham & Hebert, 2007) and UNITE (Kõljalg et al., 2013) could provide a framework against which the extent of diversity deposited at natural history collections can be measured, using DNA barcodes as quick ID tool in cases where specimens may be hard to identify. For instance, for associating ana- and telomorphs in macrofungi such quick ID can be essential. However, molecular approaches to identify hidden diversity remain debatable (e.g., Brower, 2006) but can be overcome in large clades such as Lepidoptera (Hebert, DeWaard & Landry, 2010). In UNITE fungal species hypotheses are generated and named, but also tagged with a citable digital object identifiers (DOIs). This allows unambiguous communication, and harmonisation of species concepts throughout communities. In comparison, BOLD allows barcode index numbers (BINs) to refer to barcode clusters that have not been yet described taxonomically.

Programs like the UK Darwin Initiative train observers and scientists in countries rich in diversity but low in funding for conservation and science surveys. These programs can further support democratization of not just specimens and data but also the knowledge for performing analyses and conducting research. However, much more needs to be done in this area, especially in capacity building, infrastructural development, and task distribution.

Reference collections

A major need for collections worldwide is to develop basic molecular data associated with a given taxon, but possibly genomes, transcriptomes and phenomes too. For example, nearly all museum genetic resources collections are unsuitable for gene expression work, requiring new collection of fresh samples for this endeavour (Zhang et al., 2014). The key importance is the burgeoning use of metabarcoding in ecological studies that can be anchored to museum specimens, and thus linked to the associated metadata. Many museums have embarked on such endeavors, for example at CSIRO in Australia, and efforts of multiple museums ideally come together in clade-based DNA barcode projects in BOLD.

Promoting science through museums

To enhance the broader relevance of natural history museums it is also important to message effectively to industry and policy makers. In particular, the museum community should explore ways to use specimens to find novel ways to bridge the traditional chasm between the sciences, arts and humanities. Shared themes include place-based research and experiential learning, both encouraged in instructional efforts in science, technology, engineering and mathematics (STEM fields) and the arts. Both the arts and sciences depend on inspiration, creativity, and critical assessment, and museum specimens serve well as sources for both inspiration and fascination. (The ‘Hall of Biodiversity’ at Porto’s Botanical Garden and Natural History Museum (MHNC), may serve as a prime example; see https://mhnc.up.pt/galeria-da-biodiversidade/). While scientific education and research offer rigorous methods for testing hypotheses and creating new knowledge, integration of experiential art and humanities into science fosters non-traditional ways of exploring and messaging about our world (Balengee, 2010). Natural history museums should continue their efforts to train scientists and artists to develop novel solutions to emerging problems, especially as we face an increasingly uncertain environmental future.

Efforts by a Global Museum to assemble collections that will fulfil their key roles in the future require facilitating international agreement and participation. Such a massive effort cannot remain the province of a relatively few marginally resourced programs. Identifying the answers to the most pressing questions facing society and our environment require fertile spaces for cultivating innovation in the context of training in knowledge of biodiversity. This task is impossible without museum spaces and collection resources. We cannot afford to ’waste’ the potential of natural specimens due to degradation, improper storage, or disposal, especially in the light of rapid biodiversity loss. They need space-efficient, climate-controlled and pest-free spaces. Innovations in these areas are likely needed to accommodate collections in the long term (hundreds to thousands of years) and to deal with preservation issues that may be exacerbated by global climate change (including increasing frequency of extreme weathers, hurricanes, flooding). Museum networks need to be global and there have to be initiatives and strategies developed to support and improve facilities throughout the world (see also Table 1).

Public perception of natural history museums

We argue that natural history museums should be regarded as ‘Innovation Incubators’, places where ‘anti-disciplinary’ science is thriving by building bridges between otherwise or so far improbable disciplines, and scientists from various ‘disciplines’ meet, an Academic Nexus of Integration. Because natural history collections can facilitate examining diverse aspects of taxonomic, morphological, genetic, and chemical variation across vast temporal and spatial scales, they can help diverse scientists bridge the gaps between traditional disciplines. In places where this situation is not yet in place the way to get there would be to enable ready access to both collections and research facilities, an effort that has been highly successful, for example, under the European SYNTHESYS Access scheme for the last decade. The K-12 education project “Exploring California Biodiversity” at the University of California, Berkeley (http://gk12calbio.berkeley.edu; Mitchell & Gillespie, 2007), which takes grade school students and teachers into the field, provides an excellent example of natural history museum collections broadening access and opportunities for education. Efforts such as AIM-UP! and its successor Biodiversity Literacy in Undergraduate Education (BLUE: https://www.biodiversityliteracy.com/) have combined the expertise of educators, curators, collection managers, database managers, and others in undergraduate education (Cook et al., 2014; Cook et al., 2016b; Lacey et al., 2017).

In terms of public perception of natural history museums, it is important to safeguard their role in society and justify long-term funding by continuing outreach and engaging the general public by proper messaging, for instance by initiating citizen science projects. Above, we outlined how issues regarding the justification of specimen collecting do not seem to be resolved within a vocal minority of the general public. Clearly, museum curators and scientists need to join forces in working proactively with the public to increase their awareness of, and appreciation for, the practice of rigorous biological sampling (see also Rocha et al., 2014).

Collecting for the future: integrated analysis of museum specimens for evolutionary biology

Museums need room to grow in targeted ways that will allow us to address scientific issues critical to looming societal issues such as emerging pathogens and food security (Morrison et al., 2017; Schindel & Cook, 2018) (see Box 2). Specimen-based field work should aim to preserve extensive sets of natural history material at a particular time and place that would represent multiple individuals of each species, multiple species per collecting locale, and multiple diverse aspects of individual specimens. For example, collection of mammals and their associated ectoparasites and digestive tracts has led to detailed understanding of co-evolution of hosts and parasites (Cook et al., 2017) and can fuel future studies of the role of the microbiome in such processes (Roggenbuck-Wedemayer et al., 2014; Greiman et al., 2018). Such holistic collection events can better capture the complex interactions of biotic communities and, if repeated, over time could provide key insights into changing conditions. Additionally, as mentioned above, focused, calibrated collecting of a subset of common, widespread species are necessary to provide baselines for studies of organismal response to global change. The recent attempt to estimate population numbers of North American Monarch Butterflies (Danaus plexippus) from museum collections (Boyle, Dalgleish & Puzey, 2019) met with some controversy, in part because there are clear biases and variation in collecting effort across years for this species, as well as their host plants (Riesa, Zipkin & Guralnick, 2019; Wepprich, 2019). Discussions should be held that address how we can best leverage collecting activities across the Global Museum and that planning should lead to a global effort to more rigorously inventory biodiversity.

Conclusions

For hundreds of years, natural history museums around the world have provided the general public and scientists with numerous opportunities to learn more about our natural world. Taken together, this ‘Global Museum’ must be seen as one of the most valuable assets of modern society and culture, providing the material to address challenges facing humanity today—such as baseline information against which to test hypotheses of local and global environmental change—and a critical regional cultural touchstone for the public. Natural history museums, as Academic Nexes of Integration, can function as inter-disciplinary meeting places, or innovation incubators, where questions are addressed that we did not consider asking before. The core of these institutions are the specimens. To maximise their use, it is therefore imperative to carefully consider how to best sample, preserve, handle, and store specimens in ways that not only meet today’s demands but also new, unforeseen needs. Viewing natural history museums as critical infrastructure for scientific inquiry and public understanding may help raise their profile and awareness, facilitating continued support.

The paper is the result of a three-day workshop on ‘The role of museums in modern evolutionary biology’ organized by Chalmers University of Technology and the University of Gothenburg (Sweden), under the auspices of the Gothenburg Centre for Advanced Studies (GoCAS), and held 7-9 June, 2017. Participants from Europe, the United States, South Africa and Peru were primarily based at natural history museums and botanical gardens as researchers or administrators and were chosen to represent diverse areas of museum-based science. We are very thankful to Karin Hårding, Gunnar Nyman, and Mattias Marklund for their continuous support and assistance in the program as GoCAS organizers.

Additional Information and Declarations

Competing Interests

Author Contributions

Data Availability

Julia Clarke, Scott Edwards, and Alexander Schliep are Academic Editors for PeerJ.

Freek T. Bakker and Scott V. Edwards conceived and designed the experiments, performed the experiments, analyzed the data, prepared figures and/or tables, authored or reviewed drafts of the paper, approved the final draft.

Alexandre Antonelli, Julia A. Clarke, Per G. P. Ericson, Søren Faurby, Nuno Ferrand, Magnus Gelang, Rosemary G. Gillespie, Martin Irestedt, Kennet Lundin, Ellen Larsson, Pável Matos-Maraví, Johannes Müller, Ted von Proschwitz, George K. Roderick, Alexander Schliep, Niklas Wahlberg and Mari Källersjö conceived and designed the experiments, performed the experiments, analyzed the data, authored or reviewed drafts of the paper, approved the final draft.

Joseph A. Cook conceived and designed the experiments, performed the experiments, analyzed the data, prepared figures and/or tables, authored or reviewed drafts of the paper, approved the final draft.

John Wiedenhoeft conceived and designed the experiments, performed the experiments, analyzed the data, prepared figures and/or tables, authored or reviewed drafts of the paper, approved the final draft.

The following information was supplied regarding data availability:

Raw data was not used for this article, and the figures in the article are from the literature or from project websites.

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
