# Peer review of "The Global Museum: natural history collections and the future of evolutionary science and public education"

_PeerJ, doi:10.7717/peerj.8225_

## Round 0.1 · original submission · Major Revisions

I received comments of three reviewers. All of them see merits in your literature review, but they think your manuscript needs a major revision to be accepted. The reviewers made several excellent suggestions. If you decide to revise the work, please submit a list of changes or a rebuttal against each point which is being raised when you submit the revised manuscript.

Reviewer 1 ·

Basic reporting

Manuscripts like this with multiple authors contributing are always difficult to assemble and to assess. There is plenty of value in this paper, but the challenge comes from integration and then organization and I feel like the manuscript suffers as a result (which is not unusual of such efforts). I also represent a reviewer who may be overly opiniated about certain aspects that may not be important to the authors or to readers. It is hard to assess exactly who the audience for this manuscript is (and maybe this should be discussed with even more clarity at the beginning). I would say currently, it is the broader scientific community, the educated public and the museum community in different ways throughout. That is ambitious and problematic. I would argue different sections target these different audiences and assumptions are made throughout about what readers are going to be familiar with, when they may not be. In addition, specific examples (e.g., the collecting example of the kingfisher) are not presented with enough context to be clear who the audience actually is. This can be unproductive. Below, I found myself pulling sentences out throughout the manuscript where I felt the authors were stating things in unproductive ways or straying away from what they were trying to say, or simply weren’t clear enough for the average reader to understand what they were talking about.

The biggest issue to me is that, as currently written, the “Global Collection” they envision isn’t clearly enough defined in terms of the actual human capacity and expertise needed to carry it off. This extends to the public education aspect as well. I also do not see enough of a “plan” to actually develop this internationally. There are glimmers of this here and there, but that is all and I think this is an opportunity argue for making this effort truly global.

I also think anthropocentricism is a trap we all fall into. We are not and should not be doing biodiversity science and collections-based research only for humans.

I think the issue of fossil material needs to be developed more here. Again, there are glimmers in places, but just that. Museums are the place where this material is archived.

Experimental design

No comment

Validity of the findings

See some specific comments in the general comments section below.

Additional comments

My comments come after quoted sentences below:

“and even provide solutions to climate change, global health and food security challenges.”

Statements written like this to me undervalue the role of these collections.

“In this perspective, we discuss challenges to the realization of the full potential of natural history collections and museums to serve society.”

If the collections community spends all its efforts emphasizing the value of biodiversity to humans, the planet is probably doomed.

“Through extensive exhibits and public programming and by hosting rich communities of amateurs, students, and researchers at all stages of their careers, they provide a place-based window to focus on integration of science and discovery, as well as a locus for community engagement.”

This is what could be, but it also illustrates the issues. Most museums cannot have extensive exhibits. Many do not host rich communities of people, they could, but most of these communities are small.

“These new directions include the possibility to ask new, often interdisciplinary questions in basic and applied science; inform biomimetic design; and even provide solutions to climate change, global health and food security challenges.”

Sure, but they also serve the local community and monitor locally.

“We also explore ways in which improved infrastructure will allow higher quality science and increased opportunities for interdisciplinary research and communication, as well as new uses of collections.”

Infrastructure needs expertise

“Natural history museums, which harbour extensive records of biological diversity, have always
been meeting places for scientists, amateurs, and the public.”

They harbor THE basic authoritative records of biological diversity (observations are useful, but essentially derivative data.

“As such, museums are still considered trusted resources, at a time when many other institutions are bitterly mistrusted (Foley, 2015).” Does this mean they may not be trusted in the future?

“they are also actively used and curated by professional scientists to answer pressing problems in biology and beyond”
I’d argue that this isn’t necessarily true in most museum. Hence, a major problem.

“This wealth of metadata…” Pathogens really aren’t “metadata,” they are specimens in and of themselves.

“to global change (tracking shifts in phenotype across specimens through time; Jones & Daehler, 2018)”

While I agree with this, the truth is that local change is just as important as global change and our community needs to recognize that.

“Participants, most of whom were based at natural history museums (broadly defined to also include botanical gardens; see below)”

I would not call this necessarily “broadly defined.” By this, I mean that collections-based institutions include seed banks et cet.

“Additionally, citizen science increasingly contributes to collections, which today are housed all over the world, and serve as gems of diverse global centres of cutting-edge research (see Fig. 1).”

While using “Gems” has “lustre,” they are equally factories of knowledge and they are centers of education about the natural world for an increasingly urban human population. (e.g., Bates, 2007). We always need to be on guard about communicating the true breadth of what we do.

“Natural history museums may be located at universities, sometimes without exhibits, or may include public exhibits, such as typically occurs in national, state or regional entities.”

The issue of museums and collections should probably be dealt with more directly. They may be private/public entities.

“Regionality therefore, can be considered a strength of collections and fulfills an important role in sustaining regional pride in biodiversity.”

This an anthropocentric view that I hope we can combat someday. While “regional pride” is an outcome, I see the goal as regional (and local) knowledge, active research and education, and appreciation of the value of biodiversity.

“Still, given the increasingly connected network of regional museums, the local depth provided by such regionality would be impossible to recover – if only for practical reasons - in a single, global museum, were it composed solely of physical specimens.

Because of the value of local monitoring, this should not even be the goal (and I know it isn’t). There are simply different scientific and educational reasons requiring global, regional and local levels.

“A grim example is the Brazil National Museum in Rio de Janeiro, where a fire destroyed an estimated 90% of the collections in several divisions in September 2018 (Phillips, 2018).”

This is a little too brief about what was lost and what might be done to mitigate the knowledge and data loss in such cases (and to suggest that rebuilding such collections is possible, if humans would see these institutions in the same way they see Notre Dame).

“Creating redundancies in collections, especially for extant species and genetic resource collections, is key to ensuring the longevity of these samples and associated data. Initiatives such as the Global Genome Biodiversity Network (GGBN; Droege et al., 2016) aim to collect, catalogue, and “democratise” genomic resources across global collections, covering 50,626 species (as of 18 march 2019).”

This is a weird statement and it is not clear to me how it relates to the previous examples of protecting our biodiversity knowledge base from catastrophic losses of knowledge.

186 “… of now-extinct animal species, such as the Tasmanian tiger, the golden toad, and the Hawaiian crow.”

Hawaiian Crow is not extinct. I would encourage use of scientific names throughout.

192 :These require… storage resources from those that traditionally constitute museum infrastructures, namely large scale and secure long-term storage of image data.”

This is true, but we, as a community, need to be specific, these require digital storage which is distinct from the physical, and they also require critical curation by experts.

“Genomic analyses of single bone fragments can inform on the evolutionary and demographic history of our own species (e.g., Slon et al., 2018).”

Great, but this is one in millions of species out there on the planet.

“…many collections now come from large scale ecological studies (e.g. NEON in the
US).”

I would argue it is still “some” and too often ecological research is conducted in ways that could archive specimen data, but choose not to (and of course local/regional museums often aren’t adequately staffed to effectively argue that they could assist in such efforts).

“Museum spaces ideally are filled with students who learn to think anti-disciplinarily and appreciate the importance of the specimen. These spaces can therefore be considered ‘Innovation Incubators’ where a next generation of critical thinkers in biology and beyond will be trained.”

This may be a visionary statement, but I think it misses what museums bring to the system. They bring the expertise of taxonomic groups necessary to provide context innovation incubators too. If we fill our museums with innovation incubators, they are likely to be a bunch of smart folks, but will that help biodiversity itself? I wonder if a coordinated effort that includes such innovators is what is needed.

“In addition to such observations recorded in the field, however, collected specimens (when available) offer additional options for confirming or extending the original work using new analytical techniques.”

Boakes et al. 2010 is a useful comparative example for a group of birds.

“Places are not simply a semantic convenience. It is a meaningful lens for viewing the world because it is orderly with respect to geographic space.”

Change “It” to [They]

284 “Place-based learning and education is well developed (Gruenewald and Smith, 2014) and provides a context for local understanding and societal change.”

This paragraph meant little to me after reading it several times. The following paragraph also didn’t work well for me. “Place-based” is being defined based on the specimen and where it comes from, I think. While that is one version of “Place-based.” Another if where a specimen is institutionally which allows for comparative study with specimens from other places. Again, both have critical value for biodiversity.

Line 315. Delete “some”

“For instance, for taxonomy, having the virtual, global, workbench of the Barcode Of Life Database BOLD (www.boldsystems.org; Ratnasingham & Hebert, 2007) allowed taxonomists globally to harmonise species delineations by collectively analyzing and interpreting DNA barcode patterns from global rather than regional data sets.”

I find this a fairly bold statement without much ancillary data in terms of publications provided. This is not to say the effort was not valuable. But were species delimitations really established and effectively coordinated with taxonomic efforts? To me, Barcodes almost are more like ebird records, they are snippets of data, that while useful, really go only so far in terms of helping taxonomists.

Line 339. This statement simply isn’t true. Rarity has been only one criterion. We need to stop saying things like this. In many groups of organisms (e.g., micro-organisms), we really do not even know what rarity means. With many vertebrates, we actually no longer collect rare taxa (which also may be a mistake in the long run, Rocha et al. 2014).

“In addition, assumptions about species ID based on morphology may be falsified by genetic data (DeSalle et al. 2005) - but also the reverse - revealing an unexpectedly high level of cryptic diversity in certain groups (e.g., Hebert et al., 2004).

Does this essentially highlight limitations associated with approaches that don’t necessarily include expertise (and other types of data)?

“Other ventures include the above-mentioned BOLD (with iBOL extending its coverage) which holds 6.6M barcode records across 0.29M species, many of which are commonly-occurring. Future collections should continue to expand with specimens sampled widely across biodiversity, but in addition should amass commonly-occurring species, which can serve as environmental monitors, especially when sufficient metadata is also collected.”

This statement is simply bizarre to me in relation to the paragraph above and I don’t think the numbers are really different. Yes, a lot of interesting sampling has been done with barcoding. Is 22.7 individuals per “species” really all that different than what is in specimen collections? Are there vouchers of all these bar-coded individuals? That would be valuable information if it is the case.

Line 362. While this sentence is true in the context of technology, it isn’t true in a “place-based” sense.

390 “Digitization of collections will be increasingly important in this respect; there are many valuable but undigitized collections residing in museums.”

This paragraph lauds the digitzation efforts which I support completely, but fails to acknowledge the essential need to reach out and digitize collections outside of developed countries (and to prioritize hyper-diverse groups like insects and invertebrates).

Line 394. These donations should be made not for “posterity” but for science.

Bolotov et al. (2018) could infer from freshwater pearl mussel collections that morphology has changed in time correlated to environmental alteration and climate change. Based on historical and recent specimens from extensive geographical sampling, the authors concluded that the latter may well have accelerated the population decline in pearl mussels over the last 100 years. The study underlines the importance of preserving large collections (many individuals) to enable meaningful statistical analysis of morphological measurements.

These sentences read to me like the collecting caused the declines. This needs to be reworded

Line 413. My guess is most readers unfamiliar with the community won’t rembmer what iDigBio is.

Line 414. There should be citation here. iDigBio annual reports?

417 A major question for the future is how the community should greatly expand the scope of digitized specimens.

Cite the new Biodiversity Collection Network report that looks forward specifically with respect to digitization efforts in the United States?

Line 418. “haphazardly” is a tough word for me to read in scientific paper. “Incidentally” would be better.
I think it fair to criticize the approach taken in the U.S. with respect to project-based digitization, but recognize that we could be shooting ourselves in the foot in such situations. We’ve spent a lot of money bar-coding things, and this could be criticized as well.

Line 426. But the truth in all this remains that even in well known taxonomic groups, expertise is critically needed to verify efforts to digitize data and effectively curate it.

“Knowing the limits in geographical and temporal distribution and the size of the collection, especially for the most common groups, may suffice for further scientific analysis.”

I don’t think there is enough information here for readers to understand what is being said here. It is an important point worthy of more discussion.

“can easily be supplemented by citizen science initiatives (e.g iNaturalist),”

Only true if species identification is easily made, clearly not the case for many such taxa.

Line 461. Add Beespotter (https://beespotter.org/)

Line 467. …Data are…

475 An abundance of undescribed species is only the tip of the iceberg on the amount of data undiscovered and undocumented in the world’s museums.

It is fine to point this out, but why not say what the issues are to solving this problem too?

Line 481. Fix this sentence: “In UNITE, fungal species hypotheses are generated and named, but also tagged with a citable digital object identifiers (DOIs) so THEY can be unambiguously communicated, allowing harmonisation of species concepts throughout communities.

506 ?Natural history museums should continue their efforts to train scientists and artists to develop novel solutions to emerging problems, especially as we face an increasingly uncertain environmental future.?

For me, an emerging problem identified in the paragraphs above is the lack of expertise necessary to rigorously describe diversity.

Line 509. This paragraph, to me, doesn’t go far enough in stating that museum networks need to be global and there have to be initiatives and strategies developed to support and improve facilities throughout the world.

Line 546. “This sad event,” This is far too vague to be useful to most readers. What is sad for the kingfisher? Or sad the way it played out in the media. And let’s be realistic. There probably are ways in which this could have been done differently. Also, even mentioning this event give credence to a small, but vocal group of people with viewpoints that probably do not warrant validation by our community.

Line 549. “Additionally, the public in this case did not appreciate the relative insignificance of scientific collecting as an agent of species loss as compared with habitat loss and introduced or feral predators, such as house cats.

This is overstating things in a way that doesn’t make sense to do. Some members of the public felt this way, but most didn’t, or they hadn’t or didn’t even think about it. Our community needs to always look for ways to effectively engage the vast majority of people who either haven’t thought about the issues or who support what we do, and this is not a minority.

Line 556. There could be more citations here to other examples (e.g., Rocha et al. 2014)

“Discussions should be held that address how we can best leverage collecting activities across the Global Museum and that planning should lead to a global effort to more rigorously inventory biodiversity.”

This is an ongoing discussion that will never end, and that is a good thing.

Line 594. These kinds of threads of ideas are useful but come across as disjointed in this manuscript, the blood of birds is not going to all that different from the blood of other things. I think could either be removed or deleted. At the very least it comes across as a weird way to end this paper on the Global museum.

Line 630. I hope these data are used for much more than just documenting the anthropogenic influences on biodiversity.

I would argue that space in the paper associated with genomic data, also be allocated to how these data will help understand the phenotype.

The conclusion of this manuscript recognizes that taxonomic expertise is needed, but doesn’t clearly articulate what needs to be done about this.

714. A. Zan

720. Molecular Ecology (capitalization)

Reviewer 2 ·

Basic reporting

Clear, unambiguous, professional English language used throughout. YES

Intro & background to show context. Literature well referenced & relevant. YES

Structure conforms to PeerJ standards, discipline norm, or improved for clarity. YES

Is the review of broad and cross-disciplinary interest and within the scope of the journal? YES

Has the field been reviewed recently? If so, is there a good reason for this review (different point of view, accessible to a different audience, etc.)?
To my knowledge the field has not been reviewed since all of the Systematic Agenda 2000 reports (not cited) were produced some years ago. The more recent literature is cited.

Does the Introduction adequately introduce the subject and make it clear who the
audience is/what the motivation is?
See comments

Experimental design

Most of these are not relevant
Yes, the article is relevant for Peerj

Validity of the findings

Multiple of these are not met. See comments

Additional comments

First, I want to say I liked this even though, as the authors will see, I was very frustrated at times. It is important to keep pointing out the importance of museums, especially in a rapidly changing world.

Talking about the state of museums and the science done in them, and proposing solutions and agendas goes back decades, but the main talking points and initiatives began in the 1990’s with a series of NSF-supported workshops that also generated meetings in Europe. These NSF-supported studies have continued over the years. They have been important for articulating the importance of collections and museums. I would guess that, collectively, museums may be better off now then ever before. This is not to say some are not in deeper trouble than others. And obviously there are challenges.

Museums and their health cannot be separated from their role in pushing forward science and education and in solving societal problems. Without those, there is little reason for society to support museums, except perhaps as pure entertainment. This paper does not articulate this strongly enough, because it remains the framework of the future of museums.

It is not clear to me who the audience is. Is it the general academic public rather than museum professional scientists (evolutionary biologists?) and administrators per se (and their public?). If the former, what exactly is the major thing the authors want the audience to take away from this article, other than modern museums do a lot of things they didn’t do in the past? And that they need support?

The introduction does not mention “Global Museum” and given that it appears frequently, I wondered if this is a critical theme of the paper or simply a cute handle. If it is a metaphor rather than a concrete idea of a plan, it should be so stated. If it is an objective, then so state. Right now I have no idea if it is a vision or an apparition.

line 111-118/ it might be good here to give the reader some very general stats. Local museums versus the big ones: “increasingly held in local museums”? If you are talking about “local” museums in London, Berlin, New York, Washington, then that may be true. But by “local” do you mean small museums in small towns and cities? Are collections “increasingly” held in these?
Percentage-wise that probably is not true.

And, yes citizen science is increasing, but what exactly is its impact on collections-based science, and in the growth of collections, given the multitudinous laws trying to protect biodiversity? And the example of citizen science in table 4 is nice but seems pretty trivial as citizen science in which citizens actually do science.

And stop using “global museum” until you specifically say what that is. Given that in some countries (US-especially through iDigBio) there are multiple “virtual” freely- accessible databases, one could call those global in some sense (at least for vertebrates). But this is much less true of other global collections, even the very big collections of Europe in some instances. So I have no idea what “global museum” means. Or even whether it is a reasonable expectation over the long term given the enormous costs of digitization and especially verification.

170/ I applaud GBIF but even a lot of its data on specimens are highly suspect (their IDs, locality data), especially from the species-diverse regions of the world. It is not what museums need, nor is it a model in my opinion. VertNet has much higher quality data (it goes into GBIF also).

209 et seq/ The justification for collecting is good, but not strong enough. Public ideas about collecting are very fragmentary or even nonexistent. There are multiple issues such that this perhaps needs its own section. Some issues perhaps to talk about: (1) why and how collecting should be defended in the face of increasing anti-collecting voices, (2) the need for vouchers of all types of specimen data (just don’t bleed for dna, etc as some museums do regularly) because field and museum IDs are often wrong, or there is no solid evidence for the taxon at a given locality (a voucher is needed), taxonomic changes, etc). There is a literature on all this.

How do we convince the public and policymakers about the importance of collecting. There are studies about the very low impact of collecting on populations. There are voices out there saying collecting is driving things extinct. Text about ethics? Permits? Laws? Some of this is briefly mentioned. BUT note, there are many biodiversity scientists, including evolutionary biologists, ecologists, etc, who are against collecting. Whole countries (India). Your article should speak to them also.

lines 257 et seq. It is no surprise that a database starts out using specimens, then allows dicey observational data, and the proportion of specimen data goes down. Duh? The solution is for museums to do more collecting, but that will never keep up with millions of bird watchers. At least those millions of observations can be examined statistically for outliers. I think here as elsewhere, the paragraph (or sections) mixex multiple topics. Have a separate section on collecting, separate from databases. There might be some additional self-reflection on the part of European museums as far as collecting. My impression is that US museums do far more collecting (with vouchers) than European museums, with stricter laws, more accountability etc. I would be exceedingly happy to be wrong on this. At least in the US, curators, staff and students better adhere to all the laws or they can get fired.

274/ Place-based discovery
This section following on previous discussion seems a bit out of place (no pun intended). Of course, “place-based” is the mantra for so much biodiversity/sustainability studies. But I do not quite get what is being said here. Evolutionary biology and collections are rarely “place-based” but at a larger scale. Some of this seemingly refers to specimens at a local level, and education activities at a locality; there is a mix.
At the same time, collecting at local scales is critical (think of the German insect collections and their connection to populational declines). Without collections, those studies would have been impossible.
It seems to me that if one had a section about the critical importance of collections, some of this might go there.

A lack of a common thread here is shown in lines 324-326 mixing developing museums and DNA barcoding in a single sentence.
This section needs rethinking. It is not cohesive to me.

330 The global museum

I would not call or infer BOLD to be a global museum. That is the best example? Basically a database of CO1 sequences? And the last thing it does is “harmonize” species delineations (the 2% rule??--is this being taken as scientifically serious?).
Barcoding plays a role in various kinds of studies, yes, but it is also antithetical to museum scholarship in multiple ways. There are many many papers written on this but not cited.

If you are going to articulate a vision of a global museum, then do it. There are some good ideas here, but what is the vision for linking data about phenomes and genomes across museums in such a way that it actually is synergistic to advancing answers to scientific questions? That is a vision. Online databases per se do not do that (except perhaps with distributional data). And NEON was NEVER a museum-oriented program that would “help document current biodiversity and variation of common species across the globe.” Those are “place-based” observatories that do place-based ecology; museums were envisioned to be the depository for all the common specimens collected for other studies, and museums did not want that.
Metaphorically, perhaps think about the giant network of seismic monitors as a model of how to link museums in gathering and integrating museum data (this is perhaps a wild thought I know). Again a “global museum” is not a linkage of specimen databases. That is relatively simple (think vertnet). Rather, how does one gather comparative data across taxonomic groups so that an investigator in Gothenburg can actually study specimens in Washington, D. C., from Gothenburg.

Of course, there are several billion specimens in the world, so one needs a plan. There are instances in many museums of collections having digitized specimens (anthropology, CT scans) and there are multiple ongoing national initiatives as potential models for ideas. And what about digitized literature in museums? There are lots of historical museum science journals not yet digitized.

This entire section does not articulate a vision for the “global museum”. Having local citizen science data contributes little to a “global museum” of science unless it is can be scaled globally like e-bird. But you don’t need museums for e-bird-like projects. Having citizen scientists collect all kinds of invertebrates (with no mention about how such rampant collecting can decimate populations—think egg-collecting in Europe, especially Britain) but not mentioning the pressures on museum staff to deal with this does not present the whole picture.

407/Further increasing…

This is a good section. One thing not discussed is the idea that names applied to specimens will change as science progresses. But data will remain. We do not want names applied to specimens, as contrast to names on the original labels, to remain stagnant. Good museums do indeed worry about synonymy, and this can be handled in databases, but it is also up to an outside investigator to know what they want to do. The key here is to track all historical changes. One reason why this paper could have devoted more to the promotion of taxonomy and nomenclature which are the bed-rock of information systems.

Not to mention VertNet as perhaps the best example of combining data bases and vetting them is short-sighted.

453/big clades-large collections I found this section to be a bit removed from reality. Envision trying to deal with an insect (or other invertebrate) collection with millions upon millions of specimens. It has taken decades to century to curate that. The solution to the problems is professional staff and infrastructure; then one can digitize, clear up the names, distributions and taxonomy and systematic relationships.

I really object to the notion that museum scientists can endorse DNA barcoding as a way to (line 478) “provide a good framework against which the extent of diversity deposited at natural history collections can be measured.” This forgets everything that museums stand for and what they have done through their history. DNA sequences are not taxa. If a museum has digitized its collection, even with old names (which can generally be reconciled fairly easily in many groups), then that is the “point” estimate of the taxonomic diversity in that collection. Again, this not saying that genetic data can’t be useful. But trying, again, to “harmonize” species concepts throughout communities is saying that there are no scientific concepts involved in museum specimens and diversity, just a CO1 gene tree. Museum collections are based on taxonomy, nomenclature, and rules, and scientific debates about what taxa are, as well as the hard work of thousands of scientists and staff. There is little thought in this section about the history and function of museums and certainly about what collections represent. Or how they are managed.

Reference collections (491). Surely there is a bigger vision that can be articulated here. Shouldn’t we want genomes and phenomes to be linked to each specimen? In the big picture? Taxa are comprised of individuals with variability. Surely a “metabarcode” is typological of a given taxon, a presumptive evolutionary unit.
And this paper keeps coming back to barcoding when the systematic world has effectively passed that by. Barcoding is about identification and it has a role there. But scientists are building more and more trees based on large-scale genomic data. There are efforts, not mentioned, that will have genomes of most vertebrates in a decade or so. This is what museum science represent in the future, along with a lot of other things.

496 It seems to me that this paragraph and others below need to be part of bigger questions that promote science in museums. And this should be a separate section.

If this paper is designed to tout the importance of museums and collections, then more expansive and concrete questions need to be addressed.

1. How does a vision of a global museum get realized? Who is part of this? What is the framework and cost of connectivity? I would say this will be a lot of money. OK, how will this happen given different funding models and the fact that museums are facing numerous fiscal and other challenges (not all of which were articulated strongly enough)? So how does one get policymakers at various levels, and the public, to say this is a top priority? There is a lack of realism about the will of the public here.

2. What has to be done across the very wide spectrum of museum situations and funding sources that need to happen to make a “global museum” possible. There needs to be more discussion about how museums can make themselves more necessary relative to other things the public can spend their money on. Just remember, if you do not articulate a real vision, no one will follow. In the US, astrophysicists periodically go to Congress and ask for billions (yes, billions) of dollars (for instrumentation) and they have been very successful. They are united. Biodiversity scientists are not.

3. Some other random thoughts. Evolutionary biology and public education are in the title, but surprisingly little is focused on these. Politically, I do not think “evolutionary biology” will sell a Global Museum idea. “Science” might.

I think this paper needs significant reorganization.

Reviewer 3 ·

Basic reporting

This article is said to be a literature review, but it is much more of an opinion piece than a well-done and comprehensive literature review. It is the result of a what sounds like a very interesting workshop held in 2017, and as such is woefully out of date. The authors also make a point of stating that there were contributors from the Global South at the workshop, but none of the authors of the paper is, all are from either northern Europe or the United States, with the majority from Sweden. This is not bad per se, but rather limits the scope of opinion presented.

The manuscript is relatively well-written, but is quite repetitive in parts and there are lots of what I would call “motherhood and apple pie statements” such as stating that natural history museums are places where the public and science meet – this is hardly novel, and has been the strapline of several papers recently (2018, 2019). I am also not quite sure what the authors mean when they say museums need “sustenance” (line 93) – I thought at first they meant funding, but the word funding follows, so its unclear. In line 17 they state that European museums were established with imperial ambitions, but I would argue this is the case for US museums as well – they were largely established with the expansionist program westwards. This makes me wonder how much the authors are au fait with the history of natural history collections. I am also not quite sure who the audience is for this paper. It alternately encourages and admonishes museums, but statements such as “Discussions should be held…”, “Museums should……” are not particularly helpful. Is the audience museums themselves – i.e., are these evolutionary biologists telling museums what they ought to be doing in the future (if so not a literature review, see below)? Or is the audience those funding and using museums, including the public, in order to garner support for these collections in the future? This ambiguity runs throughout the manuscript.

The literature referenced is the tip of the iceberg when related to the topic of museum collection use in other fields. The authors primarily cited their own work (I didn’t count, but was left with the overwhelming impression) and certainly are not providing a literature review that would let a reader explore just how museums and their collections have been used. As one of many examples, the discussion of the impact on collecting by public opprobrium (lines 540-550 approximately) is limited to a single example, while there is ample grey literature and blog content that could be used and referenced here. The manuscript is broad, but with the audience not clearly defined I am not sure of its ultimate impact.

Experimental design

Content is within aims and scope of PeerJ, but as stated earlier this is not a literature review – instead it is an somewhat out of date opinion paper. The literature has not been reviewed rigorously or thoroughly, many of the papers cited are the authors’ own. Several times quotes are used but not referenced, and whole bold statements of fact are not referenced (i.e., lines 367 that children looking at butterflies in drawers increases their connection with nature and appreciation of biodiversity – this may or may not be an urban myth).

The organisation is fine – the sections make sense, but the ms is very wordy and ideas repeat in different sections.

Validity of the findings

The manuscript is not really novel – and the discussion of the role of museums has moved o a long way since 2017. In Europe the DiSSCo consortium (not primarily for digitising specimens as the author state!) has been accepted into the EU’s Infrastructure Roadmap and numerous symposia and presentations have occurred that are relevant here. In the USA iDigBio is transitioning from NSF funding to a new sustainable model, with some difficulty it might be said. The SYNTHESYS project (referred to in the ms in several different ways) has entered into a new phase of funding with new goals and research aims.

I am not convinced this paper has a well-developed argument that is current in this fast-changing landscape.

Additional comments

The ms is the result of a meeting held in 2017, but much has occurred since then. It would be important to really cite more of the relevant literature, and also be up to date with developments in for example DiSSCo and SYNTHESYS in the Europe, and iDigBio in the USA. I would have appreciated a bit of analysis as well as to uses of collections – there is a lot of information on this around, and were the paper a real literature review, there would be merit in pulling this together. But as currently written this is more of an opinion piece (outdated as well) than a literature review, and has a distinct bias to northern Europe and the USA. The recent symposium volume on use of collections in the Royal Society Transactions B, for example, does a better job of citing more of the relevant literature. All these ideas are being discussed at many meetings and symposia, and I am not sure of the audience of this manuscript. Is it to admonish museums? or garner support for museums? An example of this is in lines 497-8 and lines 506-7 which contradict each other, and no examples of this kind of activity e.g., reaching out to artists) are provided. There are just too many statements of fact that are not referenced or adequately backed up.

---

## Round 0.2 · accepted · Accept

Congratulations again! Please work with our production team to get your paper published.

Reviewer 1 ·

Basic reporting

This manuscript has benefitted greatly from the review process. I want to commend the authors for their detailed consideration of the suggestions made by the reviewers. Having now read through the revised manuscript, while I may not agree with all aspects of the manuscript it makes a solid contribution to the literature on this subject.

Experimental design

Nothing to comment on

Validity of the findings

The manuscript is well written and the revised version corrects concerns raised by the reviewers.

Additional comments

The responses to the reviewers have been handled very well.